# Transcriptomic Profiling of Rectus Abdominis Muscle in Women with Gestational Diabetes-Induced Myopathy: Characterization of Pathophysiology and Potential Muscle Biomarkers of Pregnancy-Specific Urinary Incontinence

**DOI:** 10.3390/ijms232112864

**Published:** 2022-10-25

**Authors:** Fernanda Cristina Bergamo Alves, Rafael Guilen de Oliveira, David Rafael Abreu Reyes, Gabriela Azevedo Garcia, Juliana Ferreira Floriano, Raghavendra Hallur Lakshmana Shetty, Edson Assunção Mareco, Maeli Dal-Pai-Silva, Spencer Luiz Marques Payão, Fátima Pereira de Souza, Steven S. Witkin, Luis Sobrevia, Angélica Mércia Pascon Barbosa, Marilza Vieira Cunha Rudge

**Affiliations:** 1Department of Gynecology and Obstetrics, Botucatu Medical School (FMB), São Paulo State University (UNESP), Botucatu 18618-687, Brazil; 2Postgraduate Program in Materials Science and Technology (POSMAT), School of Sciences, São Paulo State University (UNESP), Bauru 17033-360, Brazil; 3Center for Biotechnology, Pravara Institute of Medical Sciences (Deemed to be University), Rahata Taluk, Ahmednagar District, Loni 413736, India; 4Environment and Regional Development Graduate Program, University of Western São Paulo (UNOESTE), Presidente Prudente 19050-680, Brazil; 5Department of Structural and Functional Biology, Institute of Biosciences, São Paulo State University (UNESP), Botucatu 18618-689, Brazil; 6Department of Genetics, Faculdade de Medicina de Marília (FAMEMA), Marília 17519-030, Brazil; 7Department of Physics, UNESP-IBILCE, São José do Rio Preto 15054-000, Brazil; 8Department of Obstetrics and Gynecology, Weill Cornell Medicine, New York, NY 10065, USA; 9Laboratory of Virology, Institute of Tropical Medicine, University of Sao Paulo Faculty of Medicine, São Paulo 05403-000, Brazil; 10Cellular and Molecular Physiology Laboratory (CMPL), Department of Obstetrics, Division of Obstetrics and Gynaecology, School of Medicine, Faculty of Medicine, Pontificia Universidad Católica de Chile, Santiago 8330024, Chile; 11Department of Physiology, Faculty of Pharmacy, Universidad de Sevilla, E-41012 Seville, Spain; 12Faculty of Medicine and Biomedical Sciences, University of Queensland, Herston, QLD 4029, Australia; 13Department of Pathology and Medical Biology, University of Groningen, 9713GZ Groningen, The Netherlands; 14Tecnologico de Monterrey, Eutra, The Institute for Obesity Research (IOR), School of Medicine and Health Sciences, Monterrey 64710, Mexico; 15Department of Physiotherapy and Occupational Therapy, School of Philosophy and Sciences, São Paulo State University (UNESP), Marilia 17525-900, Brazil

**Keywords:** gestational diabetes mellitus (GDM), pregnancy-specific urinary incontinence (PSUI), rectus abdominis muscle (RAM), transcriptomic profile, gestational diabetic-induced myopathy (GDiM)

## Abstract

Gestational diabetes mellitus (GDM) is recognized as a “window of opportunity” for the future prediction of such complications as type 2 diabetes mellitus and pelvic floor muscle disorders, including urinary incontinence and genitourinary dysfunction. Translational studies have reported that pelvic floor muscle disorders are due to a GDM-induced-myopathy (GDiM) of the pelvic floor muscle and rectus abdominis muscle (RAM). We now describe the transcriptome profiling of the RAM obtained by Cesarean section from GDM and non-GDM women with and without pregnancy-specific urinary incontinence (PSUI). We identified 650 genes in total, and the differentially expressed genes were defined by comparing three control groups to the GDM with PSUI group (GDiM). Enrichment analysis showed that GDM with PSUI was associated with decreased gene expression related to muscle structure and muscle protein synthesis, the reduced ability of muscle fibers to ameliorate muscle damage, and the altered the maintenance and generation of energy through glycogenesis. Potential genetic muscle biomarkers were validated by RT-PCR, and their relationship to the pathophysiology of the disease was verified. These findings help elucidate the molecular mechanisms of GDiM and will promote the development of innovative interventions to prevent and treat complications such as post-GDM urinary incontinence.

## 1. Introduction

Diabetic myopathy is a disease that affects individuals with type 1 (T1DM) or type 2 (T2DM) diabetes mellitus. This common complication is characterized by a failure to preserve muscle mass and function [1,2]. Gestational diabetic-induced myopathy (GDiM) is a disease that affects pregnant women with gestational diabetes mellitus (GDM) [3]. GDM, defined as hyperglycemia onset during the second or third trimester of pregnancy in women without a previous diagnosis of diabetes, can have life-long health implications [4].

GDM is one of the most common complications of pregnancy, with a prevalence ranging from 2% to 25% depending on the diagnostic criteria utilized and the study population [5,6,7,8]. The International Diabetes Federation recently reported that more than 463 million people between 20 and 79 years of age have diabetes [9]. Of the 20 million women affected by hyperglycemia during pregnancy, 84% have GDM [10].

Previously, we demonstrated that GDM causes pelvic floor muscle (PFM) dysfunction and long-term urinary incontinence [11]. Translational studies have described that this damage is caused by GDiM of the PFM and rectus abdominis muscle (RAM), as described in both pregnant women and diabetic pregnant rats [12,13,14,15]. 

GDM was an independent risk factor for pregnant-specific urinary incontinence (PSUI), a urinary incontinence that starts during pregnancy [11,16]. This scenario of atrophy and the transition from oxidative to glycolytic fiber type [17,18] results in skeletal muscle changes, such as muscle weakness, a shift in fiber type composition, increased collagen deposition, and an elevated collagen type I/III ratio [13,14,19]. The latter characteristics mimic those found in the GDiM of PFM and RAM, and are considered a profile of skeletal muscle injury induced by GDM [14,20,21] associated with a decreased quality of life [22]. Innovative intervention studies for GDiM involving exosomes, swimming exercise in diabetic pregnant rats and a review of the consequences of diabetes-induced myopathy (DiM) and their implications in rehabilitation have been described [12,22,23]. PSUI may be considered a new clinical entity linked to GDM with a direct or indirect relationship to GDiM. Furthermore, it seems to represent the first clinical symptom of long-term urinary incontinence, but this supposition requires more investigation.

Numerous studies have investigated the link between DiM and diverse cellular processes [24]. However, despite the wealth of information on muscle weakness and muscle wasting, the specific triggering events of diabetic myopathy remain unknown. More specifically, studies relating the mechanism of GDiM to GDM and PSUI remain incompletely evaluated. Risk factors of urinary incontinence during pregnancy have been less frequently studied, and the results have been inconsistent [24,25,26,27,28,29,30]. Identifying risk factors of urinary incontinence during pregnancy will inform decision-making for health care providers and pregnant women. With this knowledge, novel effective preventative strategies can be targeted during pregnancy to prevent the occurrence of urinary incontinence in late pregnancy and the postnatal period.

The vast majority of diseases and metabolic disorders are associated with dysfunction and imbalances in the complex network of gene expression and protein production needed to maintain homeostasis [31]. Omics’ research in identifying novel biomarkers and dysregulated biological pathways associated with GDM is still in an exploratory phase [32]. Genes with functional relevance to various stages of glucose control and metabolism, and insulin production and resistance, are potential targets for GDM research [33]. Gene profiling measures the expressions of thousands of genes simultaneously to assemble a global picture of cell functions. The gene expression profile can help to identify specific gene products that can be used as muscle biomarkers to facilitate the screening and early detection of various complications [34].

Tissues exhibit characteristic stable transcriptional signatures, and variations in gene expression may indicate disease candidates [35]. Module biomarkers identified from skeletal muscle transcriptome data have been reported to be high-performance indicators of T2DM classification. They have high reliability and specificity in a variety of tissues, such as the liver, heart, and beta cells, in addition to skeletal muscles [36]. Genomic-scale biological networks involving signaling molecules and pathways have been widely discussed as a model for data integration and analysis [32]. 

To date, there have been no attempts made to link GDM with RAM transcriptome sequences in women with PSUI. Further exploration is, therefore, necessary to increase the understanding of the involved complex physiological processes that occur during pregnancy. We have analyzed gene expression data to determine potential muscle biomarker genes of GDiM in women with GDM plus PSUI. Transcriptome sequencing provides a high-throughput method for studying the molecular mechanisms of GDiM.

Transcriptomic profiling of the RAM obtained by Cesarean (C) section from women with or without GDM and with and without PSUI was conducted as a means of elucidating the molecular genetic mechanism and muscle biomarkers of GDiM, and the underlining mechanisms linking GDM with persistent urinary incontinence [3]. We hypothesized that knowledge of GDiM transcriptomic profiling would have an impact on managing health, as well as in the diagnosis, prevention, and treatment of disease. The identification of a clinically relevant transcriptome marker may lead to improvements in the management of women with GDM and PSUI.

## 2. Results

The study design compared the transcriptome profiles of four groups: non-GDM continent (ND-C), non-GDM incontinent (ND-I), GDM-continent (GDM-C), and GDM-incontinent (GDM-I). The analytic methods included the following steps: data collection, DEGs (differentially expressed genes) analysis, enrichment analysis, screening potential muscle biomarkers genes, the pathway interaction network, and the validation of these muscle biomarkers. The workflow is shown in Figure 1.

### 2.1. Subject Clinical Characteristics

Participant characteristics at enrollment are listed in Table 1.

### 2.2. Gene Expression Profile of Groups

In this study, a total of 650 genes were identified. To investigate differential gene expressions between the four groups of pregnant women, we performed RNAseq analysis using DESeq2. The principal component analysis (PCA) plot, according to the transcriptome results, showed different profiles between four groups (Appendix A). Normalized counts are presented in Appendix A. The heatmap of gene expression (Figure 2) reveals a different transcription pattern for each experimental group. There was a tendency of more genes to be downregulated in the two diabetic groups, most notably in the GDM-I group.

### 2.3. Transcriptome Profiling of Non-Diabetic Continent Women (ND-C) in Comparison with Gestational Diabetic and Incontinent Women (GDM-I)

The comparison of expression profiling in RAM from ND-C and GDM-I women revealed 26 differentially expressed genes (DEGs) (all downregulated in GDM-I) and possible associations with both GDM and PSUI. A gene ontology enrichment analysis was performed to access and identify the associated biological processes (Figure 3a). Genes were analyzed using functional annotation tools such as Gene Ontology. The full list of differentially expressed genes is given in Appendix A.

All genes differentially expressed in this comparison were downregulated in the GDM-I group. When comparing the ND-C group with the GDM-I group, there was a lower level of enrichment of processes related to muscle hypertrophy (GO:0014897, GO:0014896) and muscle adaptation (GO:0014888, GO:0043500) to internal and external stimuli in the diabetic incontinent group. This was due to the minor regulation of genes such as TPM3, ATP2A2, GSN, MYH7, and PDLIM5. There was also a decrease in processes related to protein synthesis and translation (GO:0002181, GO:0006412, GO:0043043, GO:0043604). 

The protein–protein interaction network represented by all differentially expressed genes between NG-C and GDM-I is visualized in Figure 3b. There are clusters of genes related to each other in a network for the biosynthesis of collagen, muscle proteins, and the sarcoplasmic membrane. There is also a gene cluster related to protein synthesis and translation.

### 2.4. Transcriptome Profiling of Non-Diabetic Incontinent Women (ND-I) in Comparison with Gestational Diabetic and Incontinent Women (GDM-I)

A comparison of the expression profiles in RAM from ND-I and GDM-I women indicated 281 DEGs (18 down- and 263 upregulated in ND-I women). These genes are highly associated with gestational diabetes. A gene ontology enrichment analysis was performed to access and identify the associated biological process (Figure 4b). Up- and downregulated genes were analyzed using functional annotation tools such as Gene Ontology. The full list of differentially expressed genes is given in Appendix A.

The genes that were downregulated in GDiM women decreased processes related to protein regulation and protein localization in the endoplasmic reticulum (GO:1905550, GO:1900122). The genes BTF3, NACA, and RTN4 are responsible for carrying out protein synthesis and translation. Other processes related to muscle structure were also differentially regulated, such as muscle adaptation processes, atrophy, and the transition between fast and slow fibers (GO:0014733, GO:0014883, GO:0014891) by the genes ATP2A2, ACTN3, MYH7, and GSN. Processes related to the maintenance and generation of energy through glycogenesis (GO:0030388) (FBP2, PFKM, and ALDOA genes) and oxidative stress (GO:0006107) were exclusively regulated in this comparison. No significant gene enrichment was detected in the upregulated genes (18) in women with GDiM. 

The protein–protein interaction network represented by all genes differentially expressed between NG-C and GDM-I is visualized in Figure 4b. There are clusters of genes related to each other in the network for actin and muscle filament, collagen biosynthesis, electron transport chain, and ribosomal proteins.

### 2.5. Transcriptome Profiling of Gestational Diabetic Continent Women (GDM-C) in Comparison with Gestational Diabetic Incontinent Women (GDM-I)

The comparison of expression profiling in RAM from women with GDM-C and GDM-I revealed 14 differentially expressed genes (all downregulated in GDM-I) and a possible significant association with PSUI. A gene ontology enrichment analysis was performed to access and identify the involved biological processes. Genes were analyzed using functional annotation tools such as Gene Ontology. The full list of differentially expressed genes is given in Appendix A. When comparing the two diabetic groups in terms of the presence of PSUI, we found a lower enrichment of processes related to muscle structure and changes such as adaptation, contraction, and transition between fast and slow fibers (GO:0014883, GO:0014733, GO:0014888, GO:0043500, GO:0043502, GO:0006942, GO:0003012) compared to upregulated MyH7, ACTN3 and ATP2A2 genes in GDM-I. The protein–protein interaction network represented by all genes differentially expressed between GDM-C and GDM-I is visualized in Figure 5b. There were clusters of genes related to each other in a network for muscle contraction and filament sliding, and the post-transcription regulation of gene expression.

### 2.6. Screening for Muscle Biomarker Gene Candidates of Gestational Diabetic-Induced Myopathy (GDiM)

To select candidate muscle biomarker genes for GDiM, we analyzed the differentially expressed genes (up- or downregulated) in the GDM-I group in relation to the other groups (Figure 6a). Eight genes were differentially expressed in the GDM-I group in relation to the others: ATP2A2, EEF1A1, EIF1, G0S2, MYH7, NACA, TPM3 and UBC (Table 2). Coincidentally, these genes showed a downregulated expression profile when compared to women in the ND-C, ND-I, and GDM-C groups. Figure 6b shows the expression profiles (% counts RNA-seq) of the genes in each group. 

### 2.7. Validation of Common Potential Muscle Biomarker Genes by qRT-PCR

RNA-seq analysis was performed to select and validate biomarker candidate genes by qRT-PCR. Verification experiments with replicate samples were performed to confirm the DEG determinations. Although RNAseq is more accurate than previous chip-based transcriptome measuring platforms, such as microarray, biological replication is still necessary to accurately estimate the expression in each group. A total of eight genes were selected from the significantly detected DEGs in the RNA-seq analysis (FDR adjusted *p*-value < 0.01) (Figure 7). 

qRT-PCR was performed using eight genes with six biological replicates from each group (three of these replicates were from the same RNA-seq analysis and three were independently replicated in other women to identify the biomarker candidates). We analyzed a larger and more random set of women, and of the eight genes that showed a difference in RNA-seq, four genes were shown to be common muscle biomarker candidates for GDiM (G0S2, MYH7, NACA and TPM3), having exhibited a negative regulation profile in the GDM-PSUI group (Figure 7). Table 2 shows the genes listed, with the names and classifications of mechanisms in Gene Ontology.

To validate and compare the RNA-seq and qRT-PCR results of four candidate muscle biomarkers, the quantile normalization method was used to adjust for different scales of gene expression. Based on the normalized gene expression data, relative heatmaps were generated. In both the RNA-seq and qRT-PCR, there was separation between groups, and the patterns of hierarchical clustering were very similar (Figure 8). 

## 3. Discussion

This study addresses the pathophysiology of GDiM in the RAM muscle of women with GDM and PSUI as a predictor of postpartum urinary incontinence. The main findings suggest that (1) GDM with PSUI together alters the muscle structure, leading to muscle atrophy and weakness. (2) Non-diabetic women with PSUI, when compared to GDM-I women, had a greater capacity to break down glucose, increase muscle protein synthesis and stimulate the adaptation of muscle fibers, causing less muscle damage. (3) Comparing both diabetic women group, there was a lower response of muscle adaptation in diabetic incontinent women, a finding that may have arisen because of the decrease expression of, for example, ACTN2, which performs muscle adaptation and recovery. (4) Common muscle biomarker genes are related to the pathophysiology of the disease, enhancing the understanding of GDiM. The downregulated expressions of four potential muscle biomarker genes may result in GDiM in pregnancy and at long-term postpartum. All together, these findings open up a new perspective for regenerative medicine to treat DiM [37]. As regards the prevention strategies, current scientific evidence supports the recommendation to initiate or continue physical exercise in healthy pregnant women. Group exercise programs have positive effects on improving health, well-being, and social support [38]. This intervention associated with the biological markers could represent the innovation of translational medicine of GDiM and PSUI.

The ATP2a2, MYH7, TPM3, and GSN genes were downregulated in the GDM-I women using two variables—PSUI and GDM. ATP2A2 encodes SERCA2, which is an intracellular calcium pump located in the sarcoplasmic or endoplasmic reticulum of muscle cells. SERCAs were reported to be more sensitive markers than myosin isoforms in phenotypic adaptation in response to altered levels of contractile activity [39]. These proteins participate in the regulation of cytosolic calcium homeostasis and the coordination of gene expression and muscle cell function [40]. Tropomyosins (TPM3) and gelsolin (GSN) are the two actin filaments involved in muscle regulatory functions, including cell motility, cytokinesis, endocytosis, contractility, and the determination of cell shape and size. GSN encodes Ca 2+-regulated gelsolin, which is implicated in actin remodeling in cell growth and apoptosis [41].

Other processes that were downregulated in the GDM-I group compared to the non-diabetic continent group were related to the enrichment of protein synthesis. There was a decrease in processes related to protein processing and translation. Insulin is an important regulator of protein turnover in both skeletal and cardiac muscles. It is well established that insulin deficiency and insulin resistance accelerate skeletal muscle protein degradation [42]. The maintenance of skeletal muscle protein and the loss of muscle mass in diabetes, due to insulin resistance, results in damage to the intracellular signaling pathways involved in the maintenance and balance between the degradation and synthesis of new proteins. This depends on phosphorylation and the expression of new specific regulatory proteins [43,44].

Using diabetes as a variable, there was a decrease in the enrichment of processes related to protein synthesis/production and localization in the GDM-I group. The changing needs of protein production/synthesis and localization are monitored by signaling pathways in the endoplasmic reticulum, and changes caused in the MRA by GDiM can inhibit these signaling pathways in the myopathy group [45]. There was also a decrease in processes related to adaptation and muscle atrophy in the GDiM. Despite PSUI being a common variable between the two groups, it appears as if diabetes inhibits the synthesis and production of new proteins that are responsible for muscle adaptation in normal–continent women. The muscle atrophy observed in diabetic incontinent women may involve a decrease in cytoplasmic and mitochondrial protein synthesis, the latter being reflected in profound alterations in the respiratory chain [46].

In women with incontinence during pregnancy, but without diabetes, there was an increase in processes related to glucose metabolism, and an increase in the expression of genes encoding proteins that are involved, together with GLUT4, in insulin signaling and glucose metabolism, such as PFKM, FBP2, and ALDOA, compared to the GDM-I group. PFKM is a glycolytic enzyme that plays a key role in glycogen metabolism, catalyzing the conversion of fructose-6-phosphate to fructose-1,6-bisphosphate [47], and previous studies have shown that aldolase (ALDOA) has functional duality. In addition to its enzymatic activity, this protein plays a structural role in actin binding and polymerization, specifically binding to actin-containing stress fibers, and it may regulate muscle contraction [48,49]. The skeletal muscle acts as one of the main sites of protein storage and glucose disposal. A decrease in processes that damage muscle health through energy generation occurs for muscle maintenance in GDiM. This is due to muscle glycogen synthesis, which is the main pathway of general glucose metabolism and impaired glycogen synthesis, which was the major intracellular metabolic defect responsible for insulin resistance [50].

A common relationship that did not result in the enrichment of the processes, but that was visualized in the clusters formed in the interaction networks, was a decreased expression of genes (COLA3A1 and SPARC) related to collagen biosynthesis in women with GDiM compared to non-diabetic women (continent and incontinent). SPARC is a calcium-binding matricellular glycoprotein secreted by several types of cells in many organisms, and it is associated with ECM organization and remodeling, growth, cellular differentiation, wound repair, and tissue response to injury [51]. Ghanemi et al., 2019 [52], reported that the low expression of SPARC in skeletal muscle leads to an incapacity for post-fatigue recovery, actin disorders, and myofibril atrophy [53]. In our previous studies, it was shown that GDM can damage the extracellular matrix in both rats and women, also demonstrating a relationship between the damage of ECM components and the prevalence of long-term urinary incontinence [14,18,54]. The ECM is highly malleable and, consequently, its texture and physiological functions can be affected by pregnancy and insulin resistance during pregnancy. This is one of the muscular structures responsible for GDiM.

Finally, when comparing the two diabetic groups, using only PSUI as a variable, we observed differences related to muscle structure and function, such as fast and slow fiber transition, muscle adaptation, and contraction. In women with GDiM, the muscle progenitor cell population (particularly the satellite cell population—SC) can be negatively affected by the diabetic environment, and as such, is likely to contribute to the declining skeletal muscle health also seen in patients with T2DM [55]. The deduced muscle regeneration after 8 months of high-fat, high-carbohydrate feeding was attributed to a delay in myofiber maturation, rather than to SC activation or proliferation [56]. Others have shown that SCs incubated in a high-glucose medium are more likely to differentiate into adipocytes, suggesting that the myogenic capacity of SC may be affected by uncontrolled diabetes [57].

Muscle adaptation and satellite cell participation are necessary to achieve full adaptive potential, whether in growth, function, or proprioceptive coordination. The ACTN2 gene is part of this muscle adaptation process, and is responsible for encoding proteins located in the Z disk that help to anchor filaments and actin [58]. The downregulation of the ACTN2 gene in women with GDiM, in comparison to continent–diabetic women, may be linked to the development of myopathy. Previous studies have shown that the endogenous expression of ACTN2 changes in skeletal muscles in response to various cellular environments, and is linked to a diabetic hypertrophic cardiomyopathy in cardiac muscle [59].

After our initial overview of the genes and biological processes involved in GDiM, we analyzed candidate gene muscle biomarkers for this myopathy via RNA-seq and RT-PCR analysis. All candidate biomarker genes were downregulated: G0S2, MYH7, NACA, and TPM3. The skeletal muscle is composed of fibers of different types that express sets of metabolic enzymes and structures. Both MYH7 and TPM3 are essential components of skeletal muscle tissue, and encode related proteins. These two genes encode contractile proteins, all of which are highly expressed in slow-twitch fibers involved in the muscle contraction process. Chemical and biochemical alterations are associated with changes in metabolism, diabetes, and obesity, and are in accordance with the plasticity of muscle tissue. Changes in fiber type may be indicative of changes in fiber characteristics, such as existing composition [18,54].

There was a decrease in MYH7 expression—which encodes the slow myosin heavy chain (MyHC), the major isoform of MyHC in slow skeletal muscle fibers [60]—in RAM of women with GDiM, which indicates the impairment of muscle health and functionality in this group. The myopathy group also showed a downregulation of MYH7. Changes in the relative abundance of this gene are related to the contractile velocity of skeletal muscle. Previous studies have shown a positive correlation between the proportions of type I fibers in muscle and insulin sensitivity throughout the body, and human type I fibers are likely more important than type II fibers in maintaining blood pressure and glucose homeostasis in response to insulin [61,62]. Slow-twitch type-I muscle fibers are rich in mitochondria, exhibit a high oxidative capacity, and are resistant to fatigue [63].

Diabetes-induced changes in myosin expression were linked to skeletal muscle atrophy. This may have implications for the energy of contractility in skeletal muscle. Compared with diabetic cardiomyopathy, MHC-β alteration plays a role in cardiac dysfunction [64]. β-myosin is an enzyme that converts the energy of ATP hydrolysis into a mechanical force that drives contractility. In T2DM, there is a reduction in slow oxidative fibers, and consequently a reduction in muscle oxidative enzyme activity. The increase in glycolytic and oxidative enzymatic activities in individual muscle fibers is closely related to measures of long-term glycemic control, and may represent a compensatory mechanism in muscle as a function of altered glucose metabolism [65].

TPM3 is predominantly found in the slow-twitch musculature of mammals. This gene encodes a glycoprotein, a member of the tropomyosin family, that provide stability to actin filaments and regulates the access of other actin-binding proteins [66]. Tropomyosin interacts directly with actin filaments and is responsible for muscle contraction [67]. It also participates in the uptake of glucose in skeletal muscle and adipose tissue, promoting increased glucose clearance and insulin-responsivity. The downregulation of tropomyosin expression may be responsible for the loss of muscle contractility function [58]. 

Another biomarker candidate is G0S2, an important regulator of lipid metabolism. In human skeletal muscle, G0S2 inhibits adipose triglyceride lipase (ATGL) activity, playing a central role in the regulation of lipid metabolism and function. The first and rate-limiting step in skeletal muscle lipolysis is catalyzed by ATGL, and the expression and upregulation of ATGL activity causes resistance in primary skeletal muscle cells [68]. Changes in the expression of G0S2 can cause the accumulation of lipotoxic species in skeletal muscle, and the consequent impairment of insulin action [69]. Insulin plays a critical role in the regulation of lipid storage and resistance to skeletal muscle [70].

A hyperglycemic uterine environment increases the risk of long-term complications, including obesity, impaired glucose metabolism, and cardiovascular disease, in both mothers and offspring [71]. The findings of this study show that GDiM interferes with the expressions of the genes encoding proteins relevant to muscle structure and contraction function, and tissue damage repair. There are changes in the composition of MYHC, impairing insulin action and the metabolism of glucose and insulin. GDiM is associated with changes in the structure and physiology of the RAM, making recovery and repair physiologically difficult, and leading to long-term urinary incontinence.

In conclusion, the present study is, to our knowledge, the first to characterize the transcriptomic profiling of GDiM in RAM, the underlining molecular genetic mechanisms, and muscle biomarkers linking GDM to PSUI. In addition, key features of candidate muscle biomarker genes, and their relationships to changes induced by the hyperglycemic maternal environment and PSUI, are described. These findings will help disclose the molecular mechanisms of GDiM, and promote the development of innovative interventions to prevent and treat long-term GDM complications, such as post-GDM urinary incontinence.

## 4. Material and Methods

### 4.1. Study Population and Research Design

This investigation used data from the DIAMATER study, a prospective observational cohort approved by the institutional review board [3]. It was performed at the Perinatal Diabetes Research Center (PDRC) located at the University Clinical Hospital of Botucatu Medical School (UNESP), Brazil. Pregnant women included in the study were primiparous or in their second pregnancy, had undergone a planned C-section in their previous pregnancy, were between 18 and 40 years of age, and had their C-section performed in the PDRC. Exclusion criteria were pre-pregnancy UI, known T1DM or T2DM, preterm birth (<37 weeks gestation), multiple pregnancy, and any known connective tissue disorders or fetal anomalies. After informed consent was provided, RAM tissue was collected during C-section. All patients had prenatal care and delivered in our hospital.

The subjects were classified into the following 4 groups with six women per group:Non-GDM continent women (no PSUI) (ND-C);Non-GDM and incontinent women (with PSUI) (ND-UI);GDM and continent women (no PSUI) (GDM-C);GDM and incontinent women (with PSUI) (GDM-I).

The GDM diagnosis was confirmed as a fasting glucose levels of >92 mg/dL, and validated by a 75 g oral glucose tolerance test (GTT). If there were no alterations in both tests, the participant was included in the non-GDM group. All participants were asked whether they had experienced PSUI beginning with their current pregnancy. Participants who gave positive responses were identified as having PSUI, following the definition set by the International Continence Society [72].

### 4.2. Sample Preparation 

RAM samples were obtained during C-section from women who delivered healthy infants at term (>37 weeks of gestation). The RAM tissue sections for RNAseq and RT-PCR analysis were placed in tubes containing 1 mL of RNA Stabilization Reagent (QIAGEN GmbH, Hilden, Germany) and stored at 4 °C for 48 h. The reagent was removed and the RAM stored at −80 °C until RNA extraction. 

### 4.3. RNA Isolation

Total RNA extraction from RAM samples was performed using TRIzol^®^ (Qiagen, Hilden, Germany). All procedures were standardized and conducted according to the manufacturer’s protocol. Total RNA samples were quantified in a NanoDrop One spectrophotometer (Thermo Fisher, Waltham, MA, USA) and their integrity evaluated by capillary electrophoresis in a 2100 Bioanalyzer (Agilent, Santa Clara, CA, USA), with all samples having 260/280 nm and 260/230 nm ratios above 1.8 and RNA integrity numbers (RIN) > 7.0.

### 4.4. RNA-Seq

A total of 50 ng RNA was used for the generation of the RNAseq library. The sequencing was performed on a high-performance platform using HiSeq 2500 V 4.2× equipment (Illumina Inc., San Diego, CA, USA) in two channels (lanes) with a final paired reading of 100 pb.

### 4.5. Bioinformatic and Data Analysis

The quality control and filtering of raw readings from sequencing were examined using the fastqc program. Only reads with a Phred score ≥ to 28 were considered for use in the transcriptome assembly. In addition, to avoid the excessive use of computational resources, redundant readings (identified with the fastqc program and which did not show similarity with any sequence available in public databases) were also removed, using the trimmomatic program [73]. After removing low-quality and redundant reads, the final reads were mapped against the species reference genome. The sequencing results were mapped using the HISAT2 [74] program and quantified using the HTSeq algorithm [75].

The DESeq2 algorithm of the Bioconductor/R package was used to estimate the global differences in digital gene expression (DDE) between genotypes [76]. The analysis resulted in log2 fold change values and an adjusted *p*-value for each transcript detected. The genes were considered differentially expressed when log2 fold change was ≥1.5 and the adjusted *p*-value (padj) was ≤0.05.

The representation of the intensity and the expression pattern was performed using the heatmap.2/glots (3.0.1) program for R. The contigs normalized by the deseq function were grouped under supervision using the dendrogram function, and using the Pearson correlation coefficient with the average distance method for both rows and columns.

### 4.6. Pathway Enrichment Analysis

Gene Ontology (GO) functional enrichment analysis was performed to identify the overrepresented GO categories of differentially expressed genes using the Gene Ontology Consortium database (http://geneontology.org/), accessed on 1 June 2022 [77]. To perform pathway analysis, the PANTHER overrepresentation test (PANTHER version 17.0 Released 22 February 2022) was used. Statistical significance was calculated by Fisher’s exact test. Up- and downregulated genes were used to identify over-represented Gene Ontology in terms of biological processes based on the annotation. We considered the 10 most enriched terms according to the highest scores with more than three genes, and false discovery rate (FDR)-corrected *p*-values < 0.05 were considered statistically significant.

### 4.7. Protein-Protein Interactions (PPI) Networks

PPI networks were generated using the Search Tool for the Retrieval of Interacting Genes/Proteins (STRING) tool [78,79] (http://string-db.org/), accessed on 15 June 2022. We considered experiments, database, co-expression, neighborhood, and co-occurrence as active interaction sources. The minimum required interaction score was >0.700. The PPI enrichment *p*-value indicates the statistical significance provided by STRING.

### 4.8. Screening for Diabetic-Induced myopathy Biomarker Candidates

A screening approach was used for the identification of GDiM Biomarker Candidates. The Venn diagram was obtained using the InteractiVenn software (http://www.interactivenn.net), accessed on 10 June 2022 [80]. The three-way Venn diagram indicates the numbers of differentially expressed genes (up- and downregulation) when compared by the *t*-test in the GDM-I group in comparison to the other three groups (ND-C vs. GDM-I; ND-I vs. GDM-I and GDM-C vs. GDM-I). The numbers inside the intersections of circles denote the numbers of significant genes and biomarker candidates [81].

### 4.9. qRT-PCR Experiment for Technically Validating Detected DEGs from RNA-Seq Analysis and Validating Muscle Biomarkers Genes Candidates

We performed qRT-PCR using the same biological replicates of RNA-seq and also different biological replicates to validate the results of RNA-seq and biomarker candidates. Reverse transcription was performed using the High-Capacity cDNA Reverse Transcription Kit (Applied Biosystems, Waltham, MA, USA), as per the manufacturer’s instructions, and real-time PCR was performed in a StepOnePlus instrument (Applied Biosystems) using the Power SYBRTM Green PCR Master Mix (Applied Biosystems) according to the manufacturer’s protocol, in a final volume of 10 µL. Primers (Table 3) were designed using primer-BLAST [82] targeting all known transcripts of each gene, and not targeting genomic DNA, and were synthesized by Invitrogen. The final concentration of each primer was set within 100 to 400 nM based on prior optimization, achieving >80% efficiency without off-target amplification, based on melting curve analysis. Relative expression was calculated using the ΔΔCT method described by Pfaffl, 2001 [83], normalized to the mean of the endogenous controls ACTB and GAPDH (Appendix A) [84,85].

## Figures and Tables

**Figure 1 ijms-23-12864-f001:**
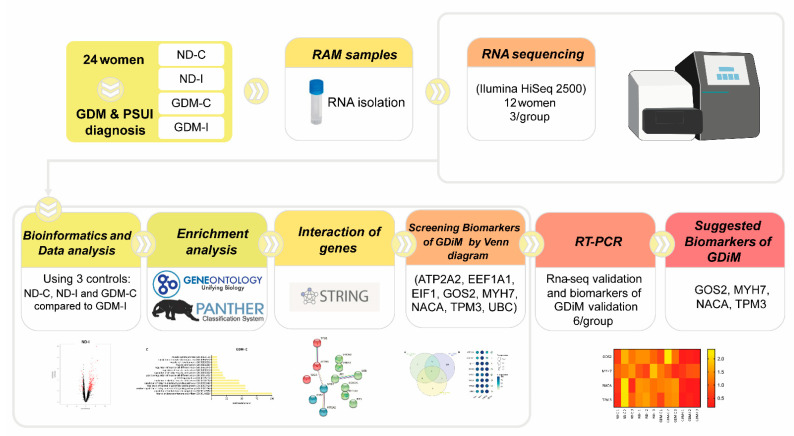
The workflow of the present study.

**Figure 2 ijms-23-12864-f002:**
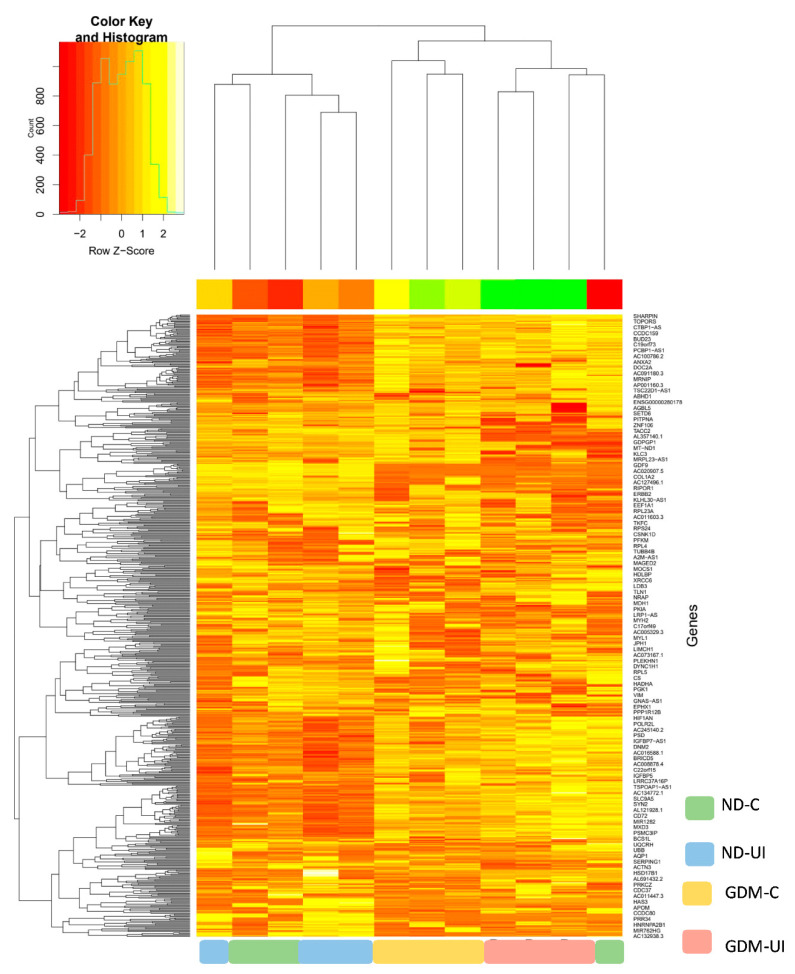
Heatmap showing relative gene expressions. The heat map depicts the relative expressions of each probe set (row) for each woman. The intensity of each block, either yellow (higher expression) or green (lower expression), represents the magnitude of difference from the mean.

**Figure 3 ijms-23-12864-f003:**
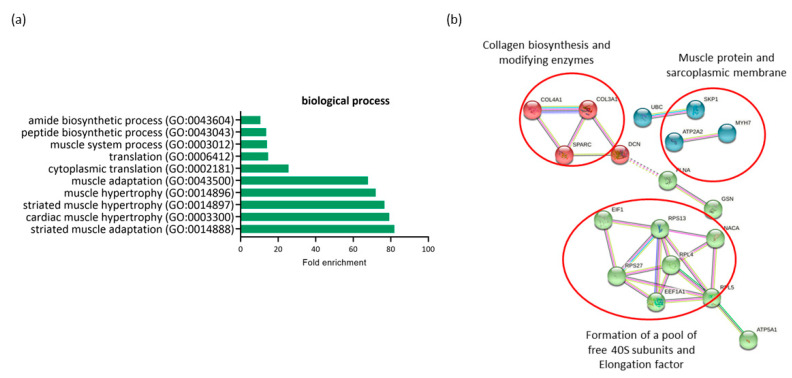
Profiling of differentially expressed genes (DEGs) of non-diabetic continent women (ND-C) in comparison with gestational diabetic and incontinent women (GDM-I). (**a**) Biological processes identified for up- and down regulated genes. Enrichment was defined as the 10 most significant terms according to the highest scores and *p*-values (<0.05). (**b**) Protein–protein interaction network representing DEGs. The interaction map was generated using STRING with clusters, a high confidence of 0.7, and all criteria for linkage (co-occurrence, co-expression, experiments, neighborhood, databases, text-mining, and homology). The genes used in this network are listed in the Appendix A.

**Figure 4 ijms-23-12864-f004:**
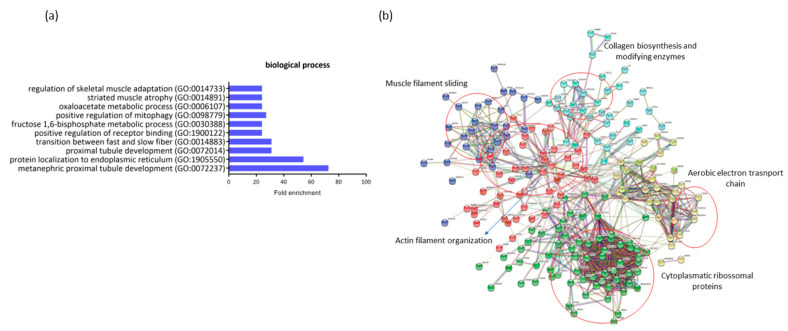
Profiling of differentially expressed genes (DEGs) of non-diabetic incontinent women (ND-I) in comparison with gestational diabetic and incontinent women (GDM-I). (**a**) Biological processes were identified for up- and downregulated genes. Enrichment was defined as the 10 most significant terms according to the highest scores and *p*-values (<0.05). (**b**) Protein–protein interaction network representing DEGs. The interaction map was generated using STRING with clusters, a high confidence of 0.7, and all criteria for linkage (co-occurrence, co-expression, experiments, neighborhood, databases, text-mining, and homology). The genes used in this network are listed in Appendix A.

**Figure 5 ijms-23-12864-f005:**
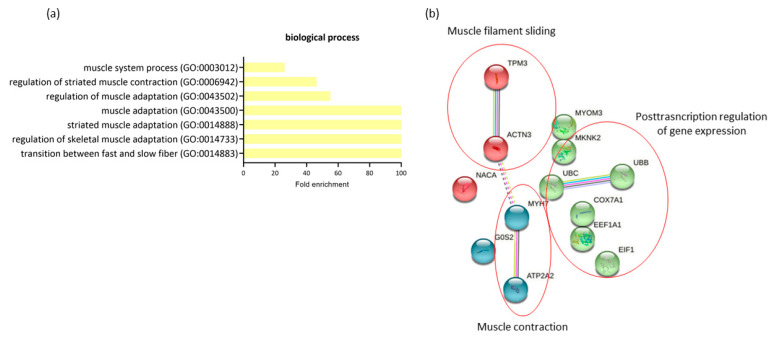
The profiling of differentially expressed genes (DEGs) of gestational diabetic continent women (GDM-C) in comparison with gestational diabetic and incontinent women (GDM-I). (**a**) Biological processes were identified for up- and downregulated genes. Enrichment was defined as the 10 most significant terms according to the highest scores and *p*-values (<0.05). (**b**) Protein–protein interaction network representing DEGs. The interaction map was generated using STRING with clusters, a high confidence of 0.7, and all criteria for linkage (co-occurrence, co-expression, experiments, neighborhood, databases, text-mining, and homology). The genes used in this network are listed in Appendix A.

**Figure 6 ijms-23-12864-f006:**
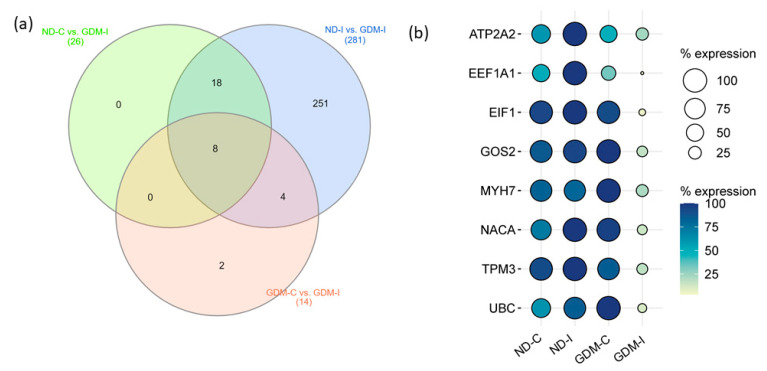
Prognostic values of potential muscle biomarker genes for gestational diabetic-induced myopathy. (**a**) Venn diagram showing the intersection of eight genes expressed differentially in the GDM-I group (GDM-I): ATP2A2, EEF1A1, EIF1, G0S2, MYH7, NACA, TPM3, and UBC. (**b**) Balloon plot showing the expression profiles of these genes, using the count/percentage of RNA-seq. ND-C: non-GDM continent women (no PSUI); ND-I: non-GDM incontinent women (with PSUI); GDM-C: GDM continent women (no PSUI); GDM-I: GDM incontinent women (with PSUI).

**Figure 7 ijms-23-12864-f007:**
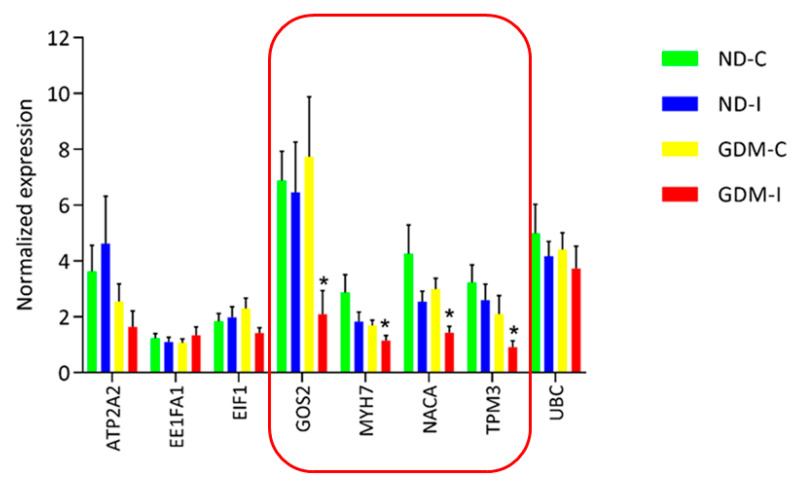
Real-time RT-PCR analysis of eight differentially expressed genes in RAM. Real-time RT-PCR was carried out on six independent biological replicates per group containing three replicates. The relative quantification of each transcript was normalized against β-actin and GAPDH. In the featured line, there are genes that were differentially expressed by RT-PCR in a larger group of analyzed women. * ND-C: non-GDM continent women (no PSUI); ND-I: non-GDM incontinent women (with PSUI); GDM-C: GDM continent women (no PSUI); GDM-I: GDM incontinent women (with PSUI).

**Figure 8 ijms-23-12864-f008:**
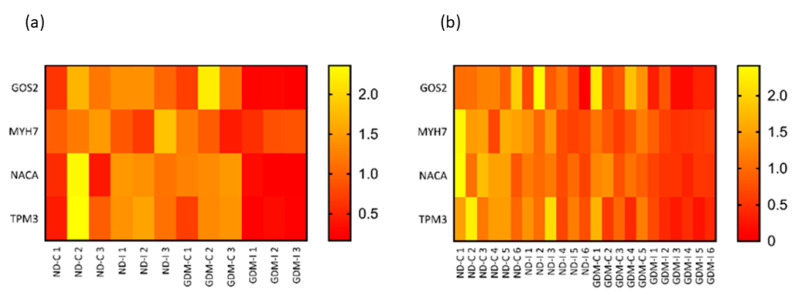
Relative heatmaps of RNA-seq (**a**) and qRT-PCR (**b**) used to validate and visualize biomarker candidates. Gene expressions from the two platforms were normalized by the quantile normalization method. ND-C: non-GDM continent women (no PSUI); ND-I: non-GDM incontinent women (with PSUI); GDM-C: GDM continent women (no PSUI); GDM-U: GDM incontinent women (with PSUI).

**Table 1 ijms-23-12864-t001:** Singleton pregnant participants recruited to the DIAMATER study cohort: baseline characteristics of women with and without GDM and PSUI.

	ND-C (*n* = 6)	ND-I (*n* = 6)	GDM-C (*n* = 6)	GDM-I (*n* = 6)
Age (years)	29.6 ± 6.40	27 ± 5.49	31 ± 4.49	28 ± 6.16
Previous C-section	0.6 ± 0.48	0.16 ± 0.4	0.8 ± 0.4	0.16 ± 0.37
Previous fetal or neonatal death	0.2 ± 0.4	0.33 ± 0.48	0	0.16 ± 0.37
BMI pre-pregnancy (kg/m^2^)	30.16 ± 9.79	29.56 ± 8.07	31.75 ± 3.15	29.03 ± 3.08
BMI delivery (kg/m^2^)	34.29 ± 9.99	34.19 ± 6.64	35.78 ± 4.04	33.11 ± 3.17
Fasting glucose (mg/dL)	66 ± 9.35	76 ± 9.31	94.2 ± 3.69 *	97 ± 11.07 *#
Oral test tolerence—1 h (mg/dL)	108 ± 13.47	129 ± 12.96	166 ± 34.30	169 ± 21.11
Oral test tolerence—2 h (mg/dL)	82 ± 22.20	115 ± 8.95	165 ± 42.23	179 ±38
Birth weight (g)	2864 ± 222	3304 ± 592	3862 ± 464 *	3750 ± 356 *

Data are expressed as means ± standard deviations. ND-C: non-GDM and continent women; ND-I: non-GDM incontinent women (with PSUI); GDM-C: GDM and continent women; GDM-I: GDM and incontinent women (with PSUI); BMI: body mass index. * *p* < 0.05—indicates significant difference compared to ND-C; # *p* < 0.05 indicates a significant difference compared to ND-I.

**Table 2 ijms-23-12864-t002:** Classification (Gene Ontology) of potential muscle biomarker genes for gestational diabetic-induced myopathy (GDiM).

Gene	Name	Biological Process	Molecular Function	Cellular Component
G0S2	G0/G1 switch protein 2	Regulation of lipid metabolic process	Glycoprotein binding	Intracellular non-membrane organelle
MYH7	Myosin-7	Muscle filament sliding	Actin binding	Muscle myosin complex
NACA	Nascent polypeptide-associated complex subunit alpha, muscle-specific form	Negative regulation of muscle cell apoptotic process	Nucleic acid binding	Nascent polypeptide-associated complex
TPM3	Tropomyosin alpha-3 chain	Muscle filament sliding	Actin binding	Muscle thin filament

**Table 3 ijms-23-12864-t003:** Primers designed using Primer-BLAST for use in real-time PCR assays.

Target mRNA	Primer Sequence		Final [ ] (nM)
MYH7	F	TCGGAGATGGCAGTCTTTGG	200
MYH7	R	TGAGGTCAAAAGGCCTGGTC	200
TPM3	F	GCACATTGCAGAAGAGGCAG	200
TPM3	R	TCTGTGCGTTCCAAGTCTCC	200
G0S2	F	GCCGTGCCACTAAGGTCATT	200
G0S2	R	GATCAGCTCCTGGACCGTTT	200
NACA	F	TCCAACTGTACAAGAGGAGAGTG	200
NACA	R	GCTTGTGACATGACCAATTCAATG	200
ATP2A2	F	AATTGCTGTTGGTGACAAAGTTCC	300
ATP2A2	R	AGTGTGCTTGATGACAGAGACAG	300
UBC	F	GGGATTTGGGTCGCAGTTCT	200
UBC	R	CTTACCAGTCAGAGTCTTCACGA	200
EEF1A1	F	TGCCAGTTCCTCGTAGAGATTG	200
EEF1A1	R	GCCACAAGCACTTAAAACCCA	200
EIF1	F	TGCCAGTTCCTCGTAGAGATTG	300
EIF1	R	GCCACAAGCACTTAAAACCCA	300
GAPDH	F	TCACCATCTTCCAGGAGCGA	400
GAPDH	R	AGCATCGCCCCACTTGATTT	400
ACTB	F	ATTGCCGACAGGATGCAGAA	200
ACTB	R	CGCTCAGGAGGAGCAATGAT	200

## Data Availability

All the produced data of the other analyses are contained within this article, or in the Appendix A.

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
