# Peer review of "Transcriptomic Profiling of Rectus Abdominis Muscle in Women with Gestational Diabetes-Induced Myopathy: Characterization of Pathophysiology and Potential Muscle Biomarkers of Pregnancy-Specific Urinary Incontinence"

_ijms, 2022, doi:10.3390/ijms232112864_

Round 1

Reviewer 1 Report

Summary statement

Alves et al. have performed transcriptome profiling in rectus abdominus (RA) muscle tissue in order to identify dysregulated genes, gene pathways, protein-protein interactions, and biomarkers that mediate the pathogenesis of gestational diabetes-induced myopathy (GDiM) of the RA and pelvic floor muscles and subsequent urinary incontinence. The four groups assessed were the following: Non-diabetic, continent; non-diabetic, incontinent; gestational diabetes, continent; and gestational diabetes, incontinent. Key genes identified in RNAseq studies and subsequent analyses were validated by qRT-PCR.

Strengths of the study

The methodology appears to be sound (i.e., RNA isolation, RNAsequencing, pathway analyses, qRT-PCR, etc.). This research brings new knowledge of the pathophysiologic processes involved in a prevalent disease state, which is an important step towards diagnosing and developing future treatments in order to improve the quality of life and overall health outcomes in this patient population.

Areas to be improved

Major points

There is only an “n” of 6 for each of the four study groups (i.e., 24 participants total). Given interpatient variability, do these sample sizes achieve statistical power?

The average BMI of all study participants both pre-pregnancy and at the time of delivery ranged from 29 to 35 kg/m2. These BMI values indicate obesity, which itself is a risk factor for pelvic floor muscle dysfunction and could confound any positive correlations made between gestational diabetes and pregnancy-specific urinary incontinence (PSUI). This issue should be addressed.

RAM biopsies were obtained from study participants who underwent Cesarean section. However, vaginal delivery more likely impacts (i.e., compromises) the pelvic floor muscle. Given that the identified biomarkers that are predictive of GDiM/PSUI are measured via a muscle biopsy, how can this diagnostic approach be leveraged from a practical perspective? Moreover, many women who develop GDiM and PSUI have vaginal deliveries rather than C-sections. How would biomarker screening be performed in this patient population? In other words, what is the clinical relevance in terms of prevention and treatment?

In the Discussion section, the authors conclude that “the down-regulated expression of eight potential myopathy biomarkers genes may result in GDiM in pregnancy and at long term postpartum.” However, only 4 of the 8 genes were down-regulated to a statistically significant degree (qRT-PCR analyses, Figure 7).

Data interpretation is not possible for most of the figures. The small font used for many of the figure scales is virtually impossible to read and should be enlarged for legibility. 

Minor points

There are grammatical and punctuation mistakes throughout the manuscript that require editing.

Sentence structure was also poor at times and should be revised. Some examples are highlighted below:

p. 3, line 125: The high value of this understudied area of diabetes research motivated development of this investigation.

p. 11, line 322: (2) Non-diabetic women with PSUI have a greater ability to break down glucose…

p. 13, line 449-450: …loss of contractility function.

p. 14, line 472: These findings are useful in disclosing the molecular mechanisms…

Could the authors please clarify the “breeding line” that is referred to on p. 10, line 308?

Author Response

Por favor, verifique o anexo

Reviewer 2 Report

The manuscript is well written and the need for the study is highlighted well. However, I think the conclusion made by the authors is not well supported by the data. I hereby have my point-by-point comments:

  1.  In the first sentence of the discussion authors state that "This study addresses the pathophysiology of GDiM in the RAM muscle of women with GDM and PSUI as a predictor of postpartum urinary incontinence". This statement is misleading as the authors did not follow up on these women whether these women have postpartum urinary incontinence or not. Thus they should edit this statement.

2. The authors utilized the genes that were downregulated in GDiM as compared to other groups as biomarkers. However, the sample was RAM, not any non-invasive or bodily fluid. Thus the use of RAM as a biomarker does not seem feasible. The authors should edit the word "biomarker". Their discovery shows the genes that might play a potential role underlying the pathophysiology of GDM-I, not as biomarkers.

3.  The authors should validate these genes at their protein level or show the activity of the respective pathway by either western blot or any other assay available, to elucidate the significance of these pathways in GDM-I. 

4. In result section 2.4, line 211- I believe ND-CI should be ND-I, and also in line 212- ND-C should be ND-I.

Author Response

Por favor, veja o anexo.

Round 2

Reviewer 2 Report

I standby with my previous comments. Authors have shown no intention of improving the data/science of the manuscript. They should remove the word biomarker as it is misleading. Their research is based on RAM tissue which is not as easily accessible for a biomarker. The manuscript has identified some genes which are involve in pathophysiology of GDiM and hence their detailed mechanism is required to determine their therapeutic role in future.

Author Response

First, We would like to thank the reviewer's comments.

We accept the changes and apologize for any acceptance that was not done in the first version. We corrected the term for muscle biomarkers, so we make it explicit now in the title and text that they are “potential muscle biomarkers”. All changes are highlighted in yellow in the text. Although we continue with the part of the term, we believe that this change helps in the understanding of the specific changes in the muscle, even the RAM collection not having as easy access to bodily fluids such as blood and urine.

             We anticipate this work as a starting point to define new potential muscle biomarkers associated with GDiM. This transcriptome profiling of skeletal muscle revealed that gestational diabetes and urinary incontinence together can be affected the muscle transcriptome.

Muscle biomarkers are commonly cited, following how they were found/in the cited articles: doi.org/10.3390/ijms23094650, doi.org/10.3390/ijms23094762. In our research, these potential muscle biomarkers are important because of every relation to the pelvic floor muscles and rectus abdominis muscle, as can be seen in our protocol of cohort study(doi.org /10.1186/s12884-020-2749-x). In the future, these studies are intended to elucidate and characterize GDiM at all levels, including postpartum women.

Regarding the mechanisms of the genes found, we changed table 2 and placed in it all the processes related to each gene. This way we have a better overview of muscle markers.

Round 3

Reviewer 2 Report

No comments